## Research Article

schools; mental health; interventions

**Corresponding author:**
Tatenda Mawoyo;
Email: mawoyo@sun.ac.za

Tatenda Mawoyo and Stefani du Toit co-lead on first authorship.

Mark Tomlinson and Sarah Skeen co-lead on senior authorship.

# Health Action in ScHools for a Thriving Adolescent Generation (HASHTAG): a feasibility trial of a school-based intervention for mental health promotion and prevention among adolescents in South Africa

Tatenda Mawoyo[1] , Stefani Du Toit[1], Christina Laurenzi[1] ,
G.J. Melendez-Torres[1,2], Mark J.D. Jordans[3,4], Nagendra Luitel[4] ,
Claire van der Westhuizen[5] , David Ross[1], Joanna Lai[6], Chiara Servili[7],
Rhiannon Evans[8], Jemma Hawkins[8], Graham Moore[8], Crick Lund[5,9],
Mark Tomlinson[1,10] and Sarah Skeen[1,11]

[1]Institute for Life Course Health Research, Department of Global Health, Stellenbosch University Faculty of Medicine and Health Sciences, South Africa; [2]University of Exeter, UK; [3]War Child, The Netherlands; [4]Transcultural Psychosocial Organization, Kathmandu, Nepal; [5]Alan J Flisher Centre for Public Mental Health, Department of Psychiatry & Mental Health, University of Cape Town, South Africa; [6]Programme Group, Health Section, UNICEF, USA; [7]Department of Mental Health and Substance Use, World Health Organization, Switzerland; [8]DECIPHer, School of Social Sciences, Cardiff University, UK; [9]Institute of Psychiatry, Psychology and Neuroscience, King's College London, UK; [10]School of Nursing and Midwifery, Queen's University Belfast, UK and [11]Mental Health, Alcohol, Substance Use and Tobacco Research Unit (MASTRU), South African Medical Research Council, Cape Town, South Africa

## Abstract

Schools play a crucial role in supporting adolescent mental health, especially in low- and middle-income countries (LMICs), where young people face structural and societal challenges. This study explores the feasibility and acceptability of the Health Action in Schools for a Thriving Adolescent Generation (HASHTAG), a multilevel intervention for at-risk adolescents aged 13–14 in South Africa. HASHTAG includes two components: thriving environment in schools (TES), a whole-school approach, and thriving together (TT), a classroom-based programme. Using a mixed-methods design, we assessed feasibility in two Khayelitsha schools through implementation measures (attendance, fidelity and acceptability), focus groups (n = 46), and pre-post surveys (n = 231). Despite COVID-19 disruptions, the intervention was implemented with high fidelity and met all progression criteria. Students and staff found HASHTAG relevant and engaging, particularly appreciating the TT sessions delivered by external facilitators. The TES teacher module also created space for reflection and self-care. Some teachers suggested improved sensitisation could strengthen the programme's impact. Although no significant changes were observed in quantitative outcomes, no harms were reported. These findings support the feasibility and acceptability of HASHTAG and highlight the need for a full-scale trial to evaluate its potential impact on adolescent mental health in LMIC settings.

## Impact statement

Adolescent mental health is an urgent public health concern, especially in low- and middle-income countries where young people face many social and structural challenges. Schools play a vital role in supporting wellbeing, yet there are few models that can be scaled and adapted to local contexts. The Health Action in Schools for a Thriving Adolescent Generation (HASHTAG) programme shows how mental health promotion can be built into everyday school life in South Africa. By combining a whole-school approach (thriving environment in schools) with a classroom-based programme (thriving together), HASHTAG connects mental health support with both school culture and learners' experiences.

Findings from the feasibility pilot study show the importance of involving adolescents as active participants through school action groups and ensuring teacher engagement to build positive school climates. The programme's strong acceptability, even during COVID-19 disruptions, shows that it can be used in schools with limited resources.

HASHTAG offers practical lessons for policymakers, educators and practitioners seeking effective ways to strengthen mental health promotion in schools. By encouraging collaboration among teachers, learners, caregivers and mental health professionals, this approach demonstrates how schools can become supportive spaces that nurture and protect adolescent mental health.

## Introduction

Adolescence, the period between 10 and 19 years, is a critical developmental phase marked by rapid physical, psychological and social changes (World Health Organization, 2021). These rapid developments can increase the risk of mental health challenges, which can be further exacerbated for individuals exposed to multiple risk factors (e.g., poverty, lack of access to health services and violence) (Knifton and Inglis, 2020). Schools are a crucial setting to address the mental health and wellbeing of adolescents (Aldridge and McChesney, 2018; Laurenzi et al., 2022). Children and adolescents with regular attendance patterns spend approximately 15,000 hours at school (Rutter, 1979; Singla et al., 2021). Schools play a central role in providing for core student needs, which include facilitating engaged learning environments, promoting healthy peer–teacher and peer–peer relationships, and recognising students' evolving autonomy (World Health Organization, 2009). They are increasingly important sites for addressing adolescent mental health (Stephan et al., 2015; Moon et al., 2017). Positive, supportive school climates can ultimately reduce the risk of mental health issues and foster a culture of wellbeing within schools (Hinze et al., 2024).

School-based psychosocial interventions have the potential to promote adolescent mental health and prevent mental disorders (World Health Organization, 2020). These interventions can be delivered by teachers, external lay counsellors or digitally (Franklin et al., 2017; Shinde et al., 2018). More school-based psychosocial interventions have begun to integrate multiple actors in an adolescent's environment – involving teachers, peers, parents/guardians and school staff (Sawyer et al., 2010; Buttigieg et al., 2015; Sanchez et al., 2017; Aldridge and McChesney, 2018). These interventions have been found to reduce emotional and behavioural problems (Burkey et al., 2018; Alozkan-Sever et al., 2023), while also providing a strong foundation for healthy behaviours and enabling adolescents to apply these behaviour changes across various aspects of their lives (Shinde et al., 2018).

While school-based programming can offer diverse options to improve a range of outcomes, implementing these interventions, particularly in low- and middle-income countries (LMICs), can be challenging, even when standard curricula and principles are applied (Zomahoun et al., 2019). Therefore, it is important to examine the acceptability and feasibility of such interventions, given the multiple barriers to delivering these interventions, such as limited school financial and human resources, and where adolescents face structural barriers such as poverty and violence.

Our team's previously published components analysis identified programme components of interventions designed to promote mental health and prevent mental disorders and risk behaviours during adolescence. We identified seven effective components across 158 universal school-based interventions: emotional regulation, stress management, problem-solving, mindfulness, interpersonal skills, assertiveness training and alcohol and drug education (Skeen et al., 2019). Critically, fewer than 10% of interventions were delivered in LMICs. While components of universal interventions are likely to support better adolescent mental health across diverse settings, more evidence from schools in LMICs is needed to understand whether and how these effects can translate to the school environment (Grande et al., 2023). Additionally, multi-level interventions integrating both session-based and school climate strategies need to be tested in LMIC settings to assess if, and how, these strategies can together shape adolescents' wellbeing (Baumann and Devkota, 2023).

Health Action in ScHools for a Thriving Adolescent Generation (HASHTAG) is a multilevel school-based health improvement intervention for young at-risk adolescents aged 13–14 years. HASHTAG was developed to address critical gaps in adolescent health and was implemented in Khayelitsha, a peri-urban community outside Cape Town, South Africa. To assess the feasibility and acceptability of HASHTAG, we conducted a feasibility trial to determine whether we could progress to a full-scale evaluation. We present here preliminary findings on HASHTAG's feasibility and acceptability in South Africa. The intervention development was led by team members from Stellenbosch University and TPO Nepal. Part of the formative work included interviews, and intervention development groups with adolescents, caregivers and teachers residing in Khayelitsha.

## Methods

### Trial design

HASHTAG was conducted in two phases. The aim of Phase 1 was to produce the HASHTAG intervention materials, integrating inputs from adolescents, parents and caregivers, teachers, school management, community stakeholders and government officials in the two study sites, South Africa and Nepal (Laurenzi et al., 2024). In Phase 2, reported in this paper, HASHTAG was implemented in schools in both sites. Using a cluster-randomised trial, we evaluated the feasibility and acceptability of HASHTAG's implementation and trial methods, providing estimates of recruitment, retention, participation rates and potential effect sizes. These findings will inform decisions on feasibility and progression to a full cluster-randomised trial of HASHTAG.

### Study aims

The study aimed to 1) determine the feasibility and acceptability of HASHTAG's two core strategies, thriving environment in schools and thriving together; 2) determine feasibility and acceptability of training, supervision and support for facilitators; 3) determine the level of fidelity of implementation of both strategies; 4) investigate barriers and facilitators to the implementation of the intervention and 5) pilot outcome measures. Progression criteria to inform likely feasibility of a future trial were based on acceptability of TES and TT strategies (≥50% of participants reporting acceptability); fidelity to strategies (≥50% of schools/country implemented the strategies with fidelity); an acceptable completion rate of measures (>80% at baseline) with adequate reliability statistics; and feasibility and acceptability of measuring outcomes at baseline and follow-up (≥75% retention rate).

### Setting

HASHTAG was conducted in Khayelitsha, South Africa and in Morang District, Nepal. This paper focuses on the South African site only. South Africa is an upper-middle-income country with high levels of socioeconomic and racial inequality. Khayelitsha is one of the largest peri-urban settlements in South Africa with high levels of crime, poverty, economic insecurity and unemployment (Edelstein and Arnott, 2019; Breetzke et al., 2021).

### School selection, matching and randomisation

Four candidate secondary schools were selected in collaboration with local government officials. We collected key indicators from each school, including number of students in school; final grade pass rate in 2020; number of educators in school; educator-to-student ratio; presence of a school governing body; whether students paid school fees or not and willingness for the school to participate in the research study. In each site, schools were numbered and paired with the most similar school, and from these pairs, a random number generator selected which school would be the intervention school. Following randomisation, we conducted meetings with school management officials at each school to inform them of their study condition and what each condition would entail. In the two intervention schools, these meetings further outlined HASHTAG, and facilitators were introduced to educators.

### Sample selection

HASHTAG was implemented with adolescents aged 13–14 years in their first year of secondary school. We aimed to recruit 60 eligible adolescents from each school (n = 240 total). We held information sessions at each school to explain the research and answer questions. Interested students received a parent consent form and provided contact details for their parent/primary caregiver. To ensure parents were fully informed about their child's participation in HASHTAG and to protect the rights of underage participants, a second consent stage was conducted. Data collectors contacted parents to confirm their awareness of the study and obtain approval for their child's participation. Subsequently, baseline assessments were scheduled with the adolescents, who reviewed an informed assent form with data collectors to ensure they understood the research purpose and their involvement. Consent was re-obtained for post-intervention assessments. Participants were informed about the voluntary nature of the study and reminded they could withdraw at any point.

A smaller subset of participants was purposively selected for post-intervention focus groups, based on willingness to participate and level of engagement. Table 1 outlines the focus group participants.

### Ethics

Ethical approval was granted by the Health Research Ethics Committee at Stellenbosch University (N19/07/088) and the Western Cape Education Department (20201106-9341).

**Table 1.** Overview of focus group participants

| Data format | Participant type | School A | School B | Total |
|---|---|---|---|---|
| Focus groups | TT participants (Grade 8 learners from intervention schools) | 8 | 5 | 13 |
| | SAG members (teacher/staff/student representations from intervention schools) | 8 | 7 | 15 |
| | Teachers that attended modules | 9 | 6 | 15 |
| | Facilitators (lay community-based facilitators that conducted the TT and TM sessions) | 2 | 1[a] | 3 |
| Grand total 46 participants | | | | |

[a]Each school had a male and female facilitator; however, one facilitator was absent during the focus group discussion.

### Intervention

#### Intervention overview

HASHTAG is based on two core strategies: a whole-school component focused on the school climate (thriving environment in schools [TES]) and a classroom-based intervention delivered directly to adolescents (thriving together [TT]). TES adopted a whole-school approach to improve school connectedness and supportive relationships and was implemented through three core activities:

1. school action groups (SAGs), comprising teachers and students in charge of implementing TES activities;
2. teacher-focused sessions, delivered over three modules covering concepts of self-care, emotional wellbeing, interpersonal skills and classroom management and
3. a mental health awareness campaign, to increase mental health literacy among students, parents and teachers.

TT was designed as a classroom-based programme delivered to adolescents by trained facilitators over six weekly 90-min sessions. Content included the core components outlined above: interpersonal skills, emotional regulation, stress management, mindfulness, problem solving, assertiveness training and alcohol and drug education (Skeen et al., 2019). A previous paper describes HASHTAG's development and session content in more detail (Laurenzi et al., 2024).

HASHTAG was delivered between June and October 2021. TES implementation was planned and executed through a series of workshops with SAGs. Teacher-focused sessions were conducted over two in-person sessions conducted offsite. Additionally, a mental health awareness campaign was launched as an art-based competition, inviting students to create artwork in their chosen medium to express their understanding of mental health. The series of lockdowns in South Africa led to school closures and subsequent alternating school schedules to enable social distancing, where students would attend 2–3 days of school weekly, instead of 5. After discussions with the school management, it was decided that TT sessions at both schools would be conducted on days when students were not scheduled to attend class. For instance, if students had classes on Monday, Wednesday and Friday, TT sessions would be conducted on Tuesday. The schedule was revised weekly to accommodate school timetable changes.

#### Control condition

After post-intervention assessments, students in the control condition schools were invited to take part in a shortened version of the classroom-based TT intervention.

#### Selection, training and supervision of HASHTAG facilitators

External facilitators implemented activities at each school in pairs. Lay facilitators from a community research centre were chosen based on their prior experience working with adolescents. They underwent a 2-week training programme covering mental health, communication skills, behaviour change, administrative management and referral mechanisms. Facilitators were trained to identify distressed adolescents and refer them using a local resource list. Each facilitator pair was assigned to one school for the intervention's duration and received weekly supervision from experienced supervisors. Supervision included reflection of the past week, role-plays of TT sessions and general facilitator support. Additionally, supervisors also conducted regular site visits during HASHTAG's

implementation. In the TT groups, each pair of facilitators managed a maximum of 31 students.

## Data collection

### Procedures

Adolescents were interviewed using quantitative questionnaires at baseline and post-intervention. Questionnaires were administered by trained, supervised data collectors who had experience working in high-adversity settings, following all necessary ethical procedures. Interview responses were captured on tablets and sent directly to a server, allowing for real-time data quality monitoring. Post-intervention focus group interviews were also conducted with adolescent participants, teachers, SAG members and intervention facilitators from intervention schools, and facilitated by a data collector. All research participants received a voucher worth R180 for each assessment or focus group discussion.

### Sample size

Because this is a feasibility trial, we did not calculate formal sample size calculations. Based on consideration of the country contexts, a sample of four schools per country was judged to be sufficient to capture within-country variation and gather lessons to apply in a future pilot trial. For the quantitative assessments, 306 participants were approached and 235 completed the baseline assessments.

## Study measures

Study measures to assess feasibility and acceptability included process evaluation measures, qualitative evaluation data and quantitative assessment measures (primary and secondary outcomes).

### Process evaluation

Process evaluation data were collected, including facilitator records of sessions, programme attendance rates and observations of participant engagement and implementer fidelity during sessions. To measure fidelity, each facilitator completed a post-session form outlining the activities completed during each session. Additionally, when supervisors attended a session, they completed an observation form and outlined all activities. A total of 76 post-session forms were completed for TT and seven post session forms were completed for the teacher-focused modules. Supervisors kept records of supervision sessions, emergent issues and actions to resolve them. Participants in the TT classroom-based sessions and teacher-focused modules were asked to provide both general and specific feedback after each session.

Focus group discussions – selected to capture diverse perspectives and enable participants to expand on each other's responses – aimed to determine the acceptability of the intervention content, structure, timing, delivery and facilitation, and probed the overall project implementation successes and challenges. Facilitators were asked about the overall student engagement over the intervention period, opinions of the training and supervision, and suggestions for improvements.

### Quantitative assessments

In structured questionnaires with adolescents, we collected sociodemographic information, including age, gender, household structure, household employment, house type and access to basic services. Indicative **primary outcomes** included positive mental health, depressive symptoms and anxiety symptoms. *Positive mental health* was measured using three scales: the Stirling Children's Wellbeing Scale (Liddle and Carter, 2015), validated among adolescents in LMIC

contexts including Bangladesh (Cronbach's alpha: 0.746) (Haque and Imran, 2016); an adapted version of the Resilience Scale (Wagnild and Young, 1993; Kohrt et al., 2016), used in similar contexts including Nigeria (Cronbach's alpha: 0.867)(Oladipo and Idemudia, 2015); and the Multidimensional Student Life Satisfaction Scale (Huebner, 1994) used in various settings including South Africa (Cronbach's alpha ranging from 0.69 to 0.94) (Abubakar et al., 2016) *Depressive symptoms* were assessed using the Patient Health Questionnaire-Adolescent version (PHQ-9-A) (Johnson et al., 2002; Haas et al., 2020). *Anxiety symptoms* were assessed using the Generalized Anxiety Disorder-7 (GAD-7) (Spitzer et al., 2006). Both the PHQ-9A and GAD-7 were validated in Khayelitsha as part of a large-scale study of mental health diagnostic measures for adolescents (Marlow et al., 2023). Cronbach's alpha from similar contexts with adolescents in the Western Cape Province of South Africa was 0.81 and 0.80, for PHQ-9-A and GAD-7, respectively (Mkhize et al., 2024). Indicative **secondary outcomes** included: *psychosocial functioning*, assessed using the Strengths and Difficulties prosocial scale (Goodman and Goodman, 2009) (Cronbach's alpha ranging from 0.57 to 0.72 in a study among South African adolescents) (Aarø et al., 2022) and the WHO Disability Assessment Schedule (Üstün et al., 2010) (Cronbach's alpha for children and adolescents in Rwanda was 0.84) (Scorza et al., 2013); s*ubstance use*, measured using the Alcohol Use Disorders Identification Test (Schmidt et al., 1995) (Cronbach's alpha was 0.978 in Lebanese adolescents) (Hallit et al., 2020) and by individual questions on tobacco and illicit drug use; *aggression*, assessed using the Aggression Scale (Orpinas and Frankowski, 2001) (Cronbach's alpha ranging from 0.77 and 0.90 in a sample of South African adolescents) (Padmanabhanunni and Gerhardt, 2019); *self-harm and suicidality*, assessed using self-report questions on self-harm intentions and behaviours; *social support*, measured by the Social Connectedness Scale (Lee and Robbins, 1995) (Cronbach's alpha was 0.848 among adolescents in Nigeria) (Edet et al., 2023) and Oslo Social Support Scale (Kocalevent et al., 2018)*; school environment*, assessed using the Beyond Blue School Climate Questionnaire (Sawyer et al., 2010) (Cronbach's alpha was 0.91 in adolescents in Bihar, India) (Shinde et al., 2018); and *bullying experiences*, assessed using the Gatehouse Bullying Scale (Bond et al., 2007) (Cronbach's alpha ranged from 0.83 to 0.90 in a study conducted at the Out Patient Department of Kafer El Dawar central Hospital with children aged 6–12 years) (Abdulaziz et al., 2024).

## Analysis

### Quantitative analyses

*Primary analyses.* Pairwise deletion was used for all primary analyses. Scale values were set to missing when individual items were missing. Between-group differences were analysed using ANCOVAs to evaluate potential harm (worsening intervention group versus static control, or static intervention versus improving control) was the primary strategy. Within-group analyses by trial arm used paired t-tests for continuous outcomes or McNemar's tests for binary outcomes. Between-group analyses included terms for intervention status and baseline covariate values. Bootstrapping (1,000 iterations, bias-corrected and accelerated confidence intervals) addressed residual non-normality and partially accounted for non-independent observations; cluster-robust standard errors were used for between-group binary outcome analyses.

*Secondary analyses.* Multiple imputation methods were evaluated for potential use in a full trial. Secondary analyses used

multiple imputation with fully conditional specifications to handle missing data. A consistent rule was applied across scales to determine whether scale scores were calculated from the mean of answered questions or set to missing. Following the application of this rule to create derived variables, multiple imputation (20 iterations) employed appropriate regression specifications for imputed variables (predictive mean matching with five nearest neighbours; logit link for binary variables), stratifying imputations by trial arm. Multiple imputation models included baseline covariates but excluded school-level terms. Subsequent analyses mirrored the primary analyses, including both within-group and between-group comparisons.

### Qualitative analyses

Due to the exploratory nature of the study, an inductive thematic analysis was used. TBM (one of the authors) thoroughly reviewed all transcripts twice, identifying emerging codes based on feasibility and acceptability. All transcripts were coded using ATLAS.ti. For quality control, CvW (one of the authors) coded 3/7 transcripts (43%), and any discrepancies were resolved. Key themes were derived from codes and clusters of codes, with broader group discussions identifying salient thematic areas and connecting these to core markers of feasibility and acceptability in line with other emerging findings.

## Results

In line with study objectives, this section addresses: 1) the feasibility and acceptability of HASHTAG's two core strategies; 2) the feasibility and acceptability of the training, supervision and support for facilitators; 3) the level of fidelity of implementation of both strategies; 4) barriers and facilitators to intervention implementation and 5) preliminary evidence of effect in pilot outcome measures.

### Feasibility and acceptability of TT and TES

We first analysed process data to ascertain feasibility and acceptability of both the classroom-based components (TT) and school-climate components (TES), using programme attendance data and post-intervention focus groups with students, teachers, SAG members and facilitators.

### Thriving together (TT)

Students and teachers appreciated HASHTAG's TT classroom-based sessions. Between the two intervention schools, mean attendance was 70%. Across both schools, 100% of participating students provided feedback on Sessions 1, 3, 4 and 6. For Sessions 2 and 5, only 2 students from School 2, and 1 student from School 1, did not provide feedback. Students' feedback was largely positive; students mostly strongly agreed that the content was interesting (87%) and presented clearly (80%), that they were made to feel welcome (82%) and that overall, the session was a positive experience (75%). Students in post-intervention focus groups gave voice to how HASHTAG exceeded their expectations (Figure 1):

> "I thought that it was going to be boring and not be taught anything, maybe do other things that are not interesting to a person, you see? But as time went by we were learning and learning, we went to the sessions once a week and realized that HASHTAG helps, there is help that it brings, so we must go all the time, go to the sessions" (TT, student focus group, School 1)

Teachers noted students' interest in HASHTAG and how they would show up on their "off" days, as rotational school schedules were still in place. This attendance also enabled teachers to have more contact time with them:

> They were also excited and keen about the program, even if it is not their day [to attend the sessions], they were coming […] So even you as a teacher you will then be able to get them for your own stuff after that program. (Teacher focus group, School 2).

According to teachers, part of HASHTAG's appeal was facilitators external to the school, who brought a fresh perspective to young participants:

> Having external facilitators [helps] for them to be removed from us, because they are getting irritated by us because when they see me, they see the subject [I teach]. However, when it is the external

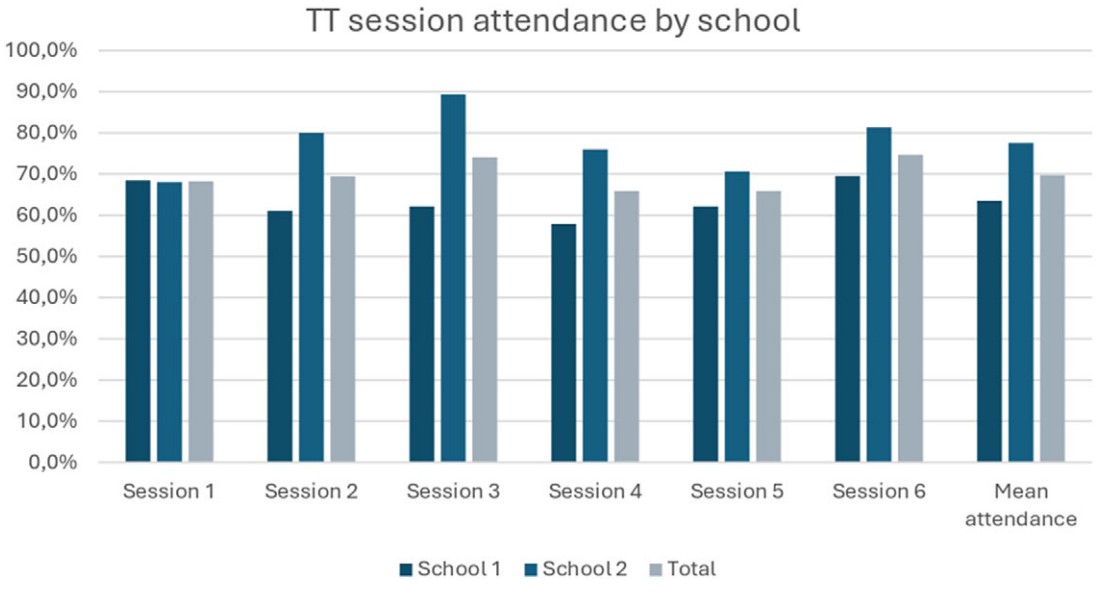

**Figure 1.** TT session attendance (August–September 2021).

facilitators coming, they are going to interact with young people, so that impresses them. (Teacher focus group, School 2)

Regarding intervention acceptability, a teacher from one of the SAGs noted that the transparent introduction of HASHTAG, and its integration without adding to teachers' burden, contributed to their accepting it:

The teachers run away from [extra programmes] because when most of the projects come, they don't come upfront like with everything […] HASHTAG, at least it came with everything upfront and it explained to us that our programme is like this […] the programme ran and it was exactly like the way they said it would, the teachers never had [a] burden. (SAG teacher, School 2)

Because of the significant burdens that teachers face in schools where students have added social, emotional and material needs, such transparency helped them to see the benefit and be clear about how it may intersect with their own commitments. However, several teachers felt that more sensitisation may have more smoothly integrated HASHTAG into their school:

Full buy-in of educators, so that we all begin to understand and see the importance of [the program] […] there is nothing much that on your side alone can do, so if you guys can just have and try to find a way of roping us in to the program, we can fully participate into the program. (Teacher focus group, School 1)

### Thriving environment in schools (TES)

Similarly, the second component, TES, was found to be acceptable. This component involved teacher modules, home-based outreach and messaging for parents/caregivers and school-wide activities.

Teacher modules, held on two Saturdays, were well-attended (77% overall attendance rate). After the success of the first joint session (Sessions 1 and 2), attendance from both schools improved for the third and final session (Table 2). Quantitative process data similar showed high response rates for session feedback among teachers (96% for Sessions 1&2, 100% for Session 3). Teachers mostly strongly agreed that content was interesting (79%), that they were made to feel welcome (77%) and that there was engagement during the session (75%), with most others selecting "agree" and few disagreements.

In post-intervention focus groups, teachers shared how having the opportunity to engage off-site was an important draw:

It was a great experience. Number 1, to begin with, just to be taken out of [our school], and to be taken to the area like [teacher module venue], immediately that was exhilarating, breathtaking, very refreshing. (Teacher focus group, School 1)

These reflections echoed some of the earlier perspectives that informed HASHTAG – speaking to both the concerns around safety in schools, as well as the immense burdens that teachers carry. Gaining distance from their usual workplace and attending a self-care focused workshop in a venue that felt special, reiterated the aims of the module. Sessions also provided opportunities for self-reflection, and reconsidering teaching practices and daily student interactions. As one teacher noted:

The issue of affirmation, try to look at good things on somebody, sometimes we focus on the negatives […] you think that you cannot learn from that person, but when we were there [in the sessions], we realized that there is a lot amongst ourselves that can make us rich. (Teachers focus group, School 1)

Beyond developing a greater capacity for self-awareness, teachers also spoke about the importance of learning self-care and integrating it into their daily lives.

We got to learn a lot of things […] sometimes we keep on pouring and pouring and we never get the chance to recharge, never get a chance to refill, you know? We got to learn that you have to know that self-care is very important, those things are things that we underestimated. (Teacher focus group, School 1)

One student also shared how intervention lessons were reinforced through parent-targeted text messaging that enabled lessons to be shared, discussed and practiced at home:

The message was saying, encourage them, make sure that they do it all the time. My mother would tell me that here is a message, it is said that you must always do mindfulness, and then she would ask me – what is mindfulness? And I would tell her what it is. And if she is not tired when she comes from work, she would say, come let's do it together and my sister […] we would do it the 3 of us. (Student focus group, School 2)

### Feasibility and acceptability of training, supervision and support

HASHTAG's training included both a series of kick-off training sessions, first initiated remotely during COVID lockdowns, and then integrated training and supervision as the facilitator team debriefed on previous sessions while simultaneously tackling content and preparing for the upcoming sessions. Facilitators described the modality of training, supervision and support as feasible and beneficial. They discussed the importance of having strong leadership within the team, and initiating training with a clear sense of the study's objectives and activities:

For me, that really helped having someone who [knows] what is this that they are talking about, and what is it that we need to do, to deliver, to know the aim of HASHTAG, and the outcomes that they are expected from the project. (Facilitators focus group)

Additionally, having an opportunity to prepare sessions and role-play potential scenarios, as well as to discuss and divide roles and tasks for each session by pair, contributed to a sense of ownership and confidence:

If ever we are going to doing [a] certain session in the following week, we will prepare for that session prior. And then when we were preparing, sometimes we got time to role-play it […] when you go through the process before you even go and implement it, it kind of makes it a bit of easier. (Facilitators focus group)

Facilitators also spoke about how initiating training remotely during COVID lockdowns was sub-optimal, because role plays and other practical activities had been missing.

COVID also affected [training], because we did most of our training over the phone, of which when we're doing it here [in-person], it was much better […] because when we are doing it here, we had enough time first and we also, we are going to roleplay. And then those roleplays were making them much easier for me, yes, than over the phone. (Facilitators focus group)

**Table 2.** Teacher-focused sessions: attendance

|  | n teachers invited | Session 1 & 2 (joint session) | Session 3 | Total |
|---|---|---|---|---|
| School 1 | *34* | 17 (50%) | 28 (82%) | 66% |
| School 2 | *42* | 33 (79%) | 39 (93%) | 86% |
| Total | 76 | 50 (66%) | 67(88%) | 77% |

Beyond aspects of content-specific training, facilitators also appreciated HASHTAG's training structure, which integrated softer skills around delivery style and self-reflection. As one facilitator noted, the role also required skills to navigate school environments and liaise closely with teachers and school management:

> There were also lessons that were incorporated in the training. Some general knowledge, some things that would help you about facilitating, about quality, because we are going to even do meetings with the teachers and everything, when we are trying to find our way in the school. So those general things on how to conduct a meeting […] how do you make sure that your session goes well? (Facilitators focus group)

### Implementation fidelity

Both the TES and TT were implemented with high fidelity. Teacher modules were delivered with 100% fidelity across all sessions, while TT maintained 100% fidelity in most sessions and 96.4% fidelity in one session. Facilitators reported high adherence to content in TT and TES teacher module sessions. SAG implementation was more variable, as groups were school-specific and had to coordinate across students, teachers and school administrators. In School 1, five members of the school-based support team, responsible for coordinating psychosocial support activities, served as teacher representatives in the SAG. This core group of teachers chose to attend the SAG sessions on a rotational basis. Student representatives included two students each from Grades 8, 9, 10 and 11. In School 2, three teachers participated, and attended consistently. Student representatives consisted of three learners from Grade 8, two from Grade 9, three from Grade 10, three from Grade 11 and one from Grade 12.

### Barriers and facilitators to implementation

In post-intervention focus groups, participants described a range of barriers and facilitators to HASHTAG's implementation. Facilitators raised implementation barriers linked to integrating into the school environment and described how mental health could feel like a competing priority with educators' responsibilities and other projects.

> You'll find that the teachers, their focus is more on their curriculum. It's more on what they want to achieve. You're not incorporating with what we want to achieve. So, it kind of made it difficult to run the project within the school and also having other projects in school. (Facilitators focus group)

However, participants reflected on the sustained momentum and interest from teachers and students alike.

> You would think that tomorrow [the students] won't come back, but the next day they return with the same energy they came with at school yesterday, that's what I enjoyed most […] sometimes it can start well, people enjoy it, [but] as it is going to an end, you see that the energy is not there. This one [HASHTAG] even at its end, even the colleagues were still interested. So, it ended with the energy that it started with. (SAG focus group, School 1)

Teachers noted how external facilitators created an important distance between "typical" learning and an engaging intervention:

> I guess it's more interesting when it's outside facilitators because they get bored with us, because they are going to think […] like in their mind, already they block, oh she is going to teach me again, so the best thing is the outside facilitators […] as teachers, we [are] part of that that is done. As they hear it, they get excited, they argue and the give their views, when it is done that way, the outside facilitators is the best one for them. (Teachers focus group, School 1)

Some teachers also felt that this separation made it somewhat challenging for them to reinforce key takeaways from HASHTAG:

> If all the teachers can be a little bit well equipped with the program, maybe before they start teaching the leaners, maybe they can have some few minutes, maybe introducing them but that will need all of the teachers to know about this. (Teachers focus group, School 1)

Importantly, COVID-19 presented logistical challenges to HASHTAG's school-based implementation. Disrupted school routines and space considerations both facilitated and complicated the delivery of HASHTAG.

> In terms of venues, because it is not easy during the COVID, all of our classrooms, they were full. [Facilitators' name], they know sometimes they used to arrive there, they would delay 10-15 minutes we are still looking for the classrooms, so then now we had to negotiate with our staff to get along to accommodate the programme. (SAG focus group, School 1)

### Outcome measures

All outcome measures were found to be feasible to implement and were completed by participants at both timepoints. Quantitative results (Table 2) showed no significant differences between intervention and control groups for either primary or secondary outcomes. Significant positive within-group effects (baseline vs. follow-up) were observed in both groups for depression symptoms, anxiety symptoms, aggression, social connection, social support and bullying. Additionally, the intervention group demonstrated a significant improvement in school climate. Importantly, although the intervention did not produce significant positive differences between the intervention and the control group, no harmful effects of a magnitude sufficient to reach significance were observed in our small sample.

### Discussion

Our study shows substantial support for the intervention across participating schools and fulfilment of the progression criteria (Tables 3 and 4), including over 50% reporting acceptability of TES and TT; ≥50% of schools implementing the intervention strategies with fidelity; >80% completion rate of measures with adequate reliability and a high retention rate between baseline and follow-up (98.3%). These data are complemented by supporting evidence from qualitative engagements and positive preliminary indications of positive effects on key outcome measures.

The context of complex social and educational challenges in Khayelitsha, South Africa, highlights the critical need for comprehensive, whole-school interventions like HASHTAG. Its multilevel approach, addressing both classroom dynamics (TT) and the broader school environment (TES), offers a promising strategy to strengthen social, psychological and environmental supports for adolescents.

Adolescent participants exposed to both TT and TES activities described how HASHTAG resonated with them. Importantly, for those participating in SAGs in the two intervention schools, these groups provided a unique platform for students to engage on equal terms with teachers, and the opportunity to discuss school-related issues. SAGs filled a critical gap in the absence of existing student leadership mechanisms and collaborative teacher-student platforms, demonstrating how opportunities for student leadership, agency and ownership of school policies support individual development and enhance school climate. School leadership shaped how

**Table 3.** Primary and secondary outcome measures

| Outcome | Between group difference (Intervention vs Control) | | | Within-group difference (Intervention) (n = 121) | | | Within-group difference (Control)(n = 110) | | |
|---|---|---|---|---|---|---|---|---|---|
| | Coef. | 95% CI Low | 95% CI High | Baseline M (SD) | Post M (SD) | Mean Diff. M (95% CI) | Baseline M (SD) | Post M (SD) | Mean Diff M (95% CI) |
| **Primary outcomes** | | | | | | | | | |
| *Positive mental health* | | | | | | | | | |
| Stirling Children's Wellbeing Scale | −0.21 | −1.62 | 1.20 | 47.20 (5.98) | 47.57 (6.32) | −0.37 (−1.50, 0.76) | 48.61 (6.04) | 48.64 (6.84) | −0,04 (−1.08, 1.0) |
| Resilience Scale | −0.12 | −0.74 | 0.49 | 25.89 (3.21) | 26.40 (3.0) | −0.50 (−1.05, 0.04) | 26.76 (2.79) | 26.92 (2.57) | −0.15 (−0.68, 0.35) |
| Multidimensional Student Life Satisfaction Scale | −0.02 | −0.75 | 0.71 | 23.88 (2.95) | 24.33 (3.01) | −0.45 (−1.01, 0.10) | 23.75 (3.74) | 24.29 (3.34) | −0.55 (−1.22, 0.13) |
| *Depression* | | | | | | | | | |
| Patient Health Questionnaire–9 (Adolescent) | −0.07 | −0.70 | 0.56 | 4.05 (3.09) | 2.83 (2.72) | 1.21 (0.57, 1.86)* | 4.28 (3.15) | 2.96 (2.44) | 1.32 (0.72, 1.91)* |
| *Anxiety symptomatology* | | | | | | | | | |
| Generalised Anxiety Disorder–7 | −0.04 | −0.54 | 0.46 | 2.70 (2.51) | 1.72 (1.93) | 0.98 (0.53, 1.43)* | 3.16 (2.67) | 1.9 (2.27) | 1.26 (0.74, 1.79)* |
| **Secondary outcomes** | | | | | | | | | |
| *Psychosocial functioning* | | | | | | | | | |
| Strengths and Difficulties Prosocial Scale | −0.01 | −0.32 | 0.30 | 7.82 (1.15) | 8.02 (1.34) | −0.19 (−0.43, 0.05) | 7.85 (1.38) | 8.04 (1.49) | −0.19 (−0.43, 0.05) |
| World Health Organisation Disability Assessment Scale −2 | −0.01 | −1.21 | 1.19 | 5.38 (5.49) | 4.60 (5.29) | 0.77 (−0.43, 1.99) | 6.15 (5.22) | 4.81 (4.61) | 1.35 (0.28, 2.41)* |
| *Substance use* | | | | | | | | | |
| Alcohol Use Disorders Identification Test | −0.10 | −0.72 | 0.52 | 2.38 (2.73) | 2.44 (2.18) | −0.05 (−0.67, 0.57) | 2.81 (3.55) | 2.53 (2.62) | 0.28 (−0.58, 1.14) |
| Self-report - tobacco and illicit drug use (OR (95% CI)) | 1.27 | 0.96 | 1.69 | | | | | | |
| *Aggression* | | | | | | | | | |
| Aggression Scale | −0.15 | −0.84 | 0.53 | 4.45 (5.18) | 2.08 (2.39) | 2.36 (1.44, 3.28)* | 4.55 (5.63) | 2.25 (3.08) | 2.29 (1.32, 3.26)* |
| *Self-harm and suicidality* | | | | | | | | | |
| Self-report - self-harm intentions and behaviours | 0.15 | −0.07 | 0.37 | 0.12 (0.74) | 0.25 (1.12) | −0.12 (−0.37, 0.12) | 0.26 (1.06) | 0.10 (0.64) | 0.15 (−0.05, 0.36) |
| *Social support* | | | | | | | | | |
| Social Connectedness Scale | −0.01 | −0.83 | 0.81 | 14.88 (3.44) | 13.20 (3.18) | 1.68 (0.91, 2.45)* | 14.88 (3.87) | 13.21 (3.57) | 1.67 (0.82, 2.53)* |
| Oslo Social Support Scale | −0.06 | −0.47 | 0.35 | 10.56 (1.86) | 11.24 (1.88) | −0.68 (−1.02, −0.34)* | 10.84 (1.87) | 11.42 (1.70) | −0.58 (−0.96, −0.21)* |
| *School environment* | | | | | | | | | |
| Beyond Blue School Climate Questionnaire | 1.37 | −0.66 | 3.39 | 70.55 (9.11) | 72.50 (8.64) | −1.95 (−3.50, −0.40)* | 72.56 (8.50) | 72.35 (9.96) | 0.21 (−1.32, 1.74) |
| *Bullying experiences* | | | | | | | | | |
| Gatehouse Bullying Scale | 0.06 | −0.03 | 0.15 | 0.29 (0.45) | 0.21 (0.39) | 0.07 (−0.001, 0.15)* | 0.32 (0.53) | 0.16 (0.32) | 0.16 (0.05, 0.26)* |

*$p < 0.05$.

SAGs were able to be established in each school; teachers with more availability were able to commit to more consistent attendance, whereas the second school opted for a rotational schedule. Both options enabled SAGs to continue, but further iterations of HASH-TAG may require closer attention to mentorship and sustained commitment to be optimally effective. Other research has shown how opportunities for student leadership, and a sense of agency and ownership of school policies, can support individual development while further enhancing the quality of the school climate (Thapa et al., 2013; Voight, 2015).

All selected primary and secondary outcome measures were feasible to implement and had acceptable completion rates. Although

**Table 4.** Progression criteria

| Progression criteria | Matrix | Result |
|---|---|---|
| Acceptability of TES and TT strategies | ≥50% of participants reporting acceptability | More than 50% of participants reported acceptability in the focus group discussions: acceptability of external facilitators, acceptability of external venue (TM) |
| Fidelity to strategies | ≥50% of schools/country implemented the strategies with fidelity | TT and TM implemented with high fidelity of 98.2% and 100% respectively |
| An acceptable completion rate of measures | >80% at baseline | All measures were 100% completed at baseline – every participant completed the questionnaire |
| Feasibility and acceptability of measuring outcomes at baseline and follow-up | ≥75% retention rate | Follow up rate of 98.3% |

no significant between-group differences were observed between the HASHTAG and control arms, this result is not unexpected in small, underpowered feasibility trials. Although HASHTAG engaged parents and primary caregivers of adolescents, this process was less extensive than planned. The COVID-19 pandemic significantly hampered the ability to fully engage this stakeholder group. However, prior research in Khayelitsha with parents and primary caregivers reiterates the importance of parenting during this pivotal developmental stage of early adolescence (Worthman et al., 2016). Caregivers in this study noted the importance of protecting their children's developmental potential, while supporting them to manage emergent context-specific risks during this period to guide them towards healthier lifelong trajectories. While HASHTAG was able to engage caregivers, the COVID-19 pandemic prevented extensive parental and caregiver engagement, and limited the extent to which we were able to ascertain the influence of caregivers on participants' school experiences and mental health.

The implementation of HASHTAG revealed valuable insights into teacher engagement and support. Part of our feasibility testing included exploring the opportunities for working with teachers and considering the competing demands that they face in their professional environments. The need for teacher buy-in has been established, including the need for consistency and sustained leadership, in introducing new initiatives at school level (Macy and Wheeler, 2020). While teacher-focused modules were highly valued and popular, introducing support for teachers at an earlier juncture in the intervention cycle could have fostered a greater sense of inclusion and contributed to more teacher excitement prior to implementation. Additionally, our findings emphasise the need for more sustained linkages to specifically selected key personnel in school settings, to enable smooth flow of communication and planning across phases.

Other interventions have highlighted the value of engaging teachers and school personnel (Manla, 2021). Our curriculum pulled from WarChild's CORE for Teachers curriculum (2021), which has been systematically implemented across multiple conflict settings. Given the extent of safety concerns and challenging home environments for students in Khayelitsha, continuing to draw from similar curricula and integrate trauma-informed approaches for teachers and administrators in similar settings shows promise (Luthar and Mendes, 2020). These considerations could be critical for refining HASHTAG prior to a larger, more robust effectiveness trial.

Finally, our findings point to key environmental challenges and opportunities for school-based, multi-level interventions in low-resource settings. As the HASHTAG intervention was planned for implementation in 2021, the COVID-19 pandemic and associated school closures introduced challenges that layered atop existing crises of student safety, resource limitations and teacher burnout (Zar et al., 2020; Kutywayo et al., 2022). School attendance, already variable, was even more inconsistent with social distancing timetables that reduced the amount of classroom time by 50%. In the wake of the pandemic, and as longer-term repercussions of school closures have become clearer, it is critical to consider how school-based interventions to support improved mental health can be best implemented in settings with multiple, concurrent crises and challenges (Kutywayo et al., 2022).

The longer-term sustainability of HASHTAG necessitates integrating its self-governing mechanisms into the school structure itself. Ensuring the smooth transition and integration within existing school systems is crucial for the ongoing success and broader impact of the programme (Zar et al., 2020). Prior research has identified principal and teacher buy-in as a critical factor in the effective implementation of school-based interventions. Accordingly, initiating HASHTAG with teacher-based modules could support more successful integration of the intervention's other components. Furthermore, regular engagements with teachers are deemed essential to reinforce key messages and sustain the intervention's objectives.

## Limitations

This study had several limitations. The findings are based on data from two schools in Khayelitsha, which means they cannot be generalised to other contexts. Nevertheless, the insights gained from this feasibility study may prove valuable in different settings, such as promoting active adolescent engagement and supporting teacher well-being. Additionally, caregiver involvement was limited during this study, despite caregivers being a crucial part of the school community. Future interventions could enhance caregiver participation by utilising existing platforms like parenting meetings.

## Conclusion

Overall, our findings indicate that HASHTAG is feasible, acceptable and shows considerable promise as a multilevel school-based mental health intervention for at-risk adolescents aged 13–14 years – and that it warrants testing in a future trial. We identified high fidelity in implementation, and overall positive feedback from students, teachers and facilitators across multiple sources of data. A refined HASHTAG intervention could benefit from strengthening parental and caregiver engagement, and adapting its structure for even greater resilience to environmental challenges. Building on practices tested in this feasibility study, including well-implemented training and supervision strategies, next steps

should include a full-scale effectiveness trial to evaluate its potential for improving mental health in this important population.

**Open peer review.** To view the open peer review materials for this article, please visit http://doi.org/10.1017/gmh.2026.10149.

**Data availability statement.** The data that support the findings of this study are available from the corresponding author, [TM], upon reasonable request.

**Author contribution.** Tatenda Mawoyo: Project managed, supervised, conducted formal analyses, investigations, contributed to methodology and manuscript drafting and reviewing. Stefani Du Toit: Participated in investigations, led the development of the intervention, methodology design, project administration, supervision and contributed to writing and editing the manuscript. Christina A. Laurenzi: Performed formal analysis, oversaw project administration and supervision and contributed to both the initial draft and subsequent revisions of the manuscript. Mark J.D. Jordans: Contributed to conceptualisation, funding, investigations, methodology and manuscript drafting. Nagendra P. Luitel: Involved in conceptualisation, data management, funding acquisition, investigations, project oversight, supervision and drafting the manuscript. Claire Van Der Westhuizen: Responsible for conceptualisation, funding, investigations, validation and drafting the manuscript. David A. Ross: Engaged in conceptualisation, funding acquisition, investigations, methodology and manuscript writing and editing. Joanna Lai: Contributed to conceptualisation, investigations and manuscript drafting. Chiara Servili: Contributed to conceptualisation, investigations and manuscript drafting. Rhiannon Evans: Contributed to conceptualisation, investigations, methodology and manuscript drafting and review. Jemma Hawkins: Conducted investigations, contributed to methodology and participated in writing and editing the manuscript. Graham Moore: Secured funding, conducted investigations, contributed to methodology and participated in writing and editing the manuscript. Crick Lund: Participated in conceptualisation, funding acquisition and manuscript drafting. Mark Tomlinson: Participated in conceptualisation, funding, investigations, resource provision and manuscript drafting. Sarah Skeen: Led conceptualisation, formal analysis, funding acquisition, investigations, methodology, project administration, supervision and contributed extensively to manuscript drafting and review.

**Financial support.** UK Medical Research Council, Grant. RE, JH and GM work at The Centre for Development, Evaluation, Complexity and Implementation in Public Health Improvement (DECIPHer), which is funded by the Welsh Government, through Health and Care Research Wales.

**Competing interests.** The authors state no competing interests.

**Ethics approval.** Stellenbosch University Health Research Ethics Committee (N19/07/088); Queens University Belfast Research Ethics Committee (MTomlinson. SREC_July19_V1), Nepal Health Research Council (342/2020 P); Department of Basic Education of the Western Cape, South Africa (20201106-9341).

**Consent to participate.** We obtained informed consent or assent from all participants.

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
