## [Reviewer Report]

Overall assessment:

This is a well-conducted feasibility trial of a school-based intervention for adolescent mental health in South Africa. The rationale is strong, the design is rigorous, and the findings are highly relevant to global mental health, particularly in LMIC contexts. The paper is well-structured, clear, and comprehensive. Importantly, the study contributes valuable evidence on feasibility and acceptability of multi-level school interventions.

Strengths:

1.Strong justification for the intervention, addressing a critical evidence gap in LMIC school-based programs.

2.Comprehensive mixed-methods design integrating quantitative and qualitative data.

3.High implementation fidelity despite COVID-19 disruptions, adding credibility to feasibility.

4. Inclusion of multiple stakeholders (students, teachers, facilitators, caregivers).

5. Clear description of intervention components (TES and TT), training, and supervision.

6.Transparent reporting of null findings on quantitative outcomes, with emphasis on feasibility and next steps.

Points for minor revision:

1. Clarity on progression criteria: While these are listed, it would strengthen the manuscript to explicitly show in the Discussion how each criterion was met (rather than scattered across Results/Discussion).

2. Parental/caregiver engagement: The limitations section could further emphasize the shortfall in parental engagement and outline strategies for better integration in future iterations.

3. Figures and tables: The manuscript currently includes placeholders (“[insert Table/Figure]”). These must be added, formatted, and referenced correctly.

4. Generalisability: Since the findings are from only two schools in Khayelitsha, consider explicitly acknowledging limits to external validity while still arguing for importance in similar LMIC contexts.

5. Minor language edits: A light proofread would help polish phrasing (e.g., occasional long sentences could be streamlined).

Conclusion:

This is an important and timely contribution to the literature on adolescent mental health interventions in LMICs. With minor revisions for clarity and completeness, it will make a valuable publication in Cambridge Prisms: Global Mental Health.

---

## [Reviewer Report]

This is a valuable and well-conceived study that makes a meaningful contribution to the understudied area of mental health promotion and prevention in low- and middle-income countries (LMICs), particularly within the African context. The authors address an important gap by piloting a school-based intervention aimed at supporting adolescent mental health—an area with limited empirical evidence despite its pressing need. The study has strong potential to advance understanding and inform future programming and research in similar contexts. I found the work highly promising and have provided several recommendations below to help further strengthen the manuscript and enhance its clarity and impact.

Impact Statement

1. The current impact statement reads more like an expanded abstract, summarizing background, methods, results, and recommendations. However, the purpose of an impact statement is different from that of an abstract—it should concisely convey the broader significance, contribution, and potential real-world influence of the research rather than summarizing the study design and findings. Consider focusing on how this intervention contributes to improving mental health outcomes in schools, advancing evidence-based practice, or shaping education and health policy. I would recommend the authors:

- Focus less on procedural or methodological details and more on why the findings matter, for example, how the intervention could inform school-based mental health policy, programming, or practice.

- Highlight the societal or policy relevance, innovation, or unique contribution of their work.

- Articulate how the research could influence stakeholders such as educators, practitioners, or policymakers.

- Keep the tone forward-looking, emphasizing impact and applicability rather than describing study steps or suggesting “further studies.”

Introduction

1. Last paragraph under introduction section, the authors note that the intervention was developed to address critical gaps in LMIC settings. However, it is unclear who developed this intervention. Providing context about the developers or originating institution/organization is important because it helps establish the credibility, theoretical grounding, and contextual relevance of the intervention. It also allows readers to understand whether this is a locally developed program tailored to the setting, or an adaptation of an existing intervention from another context.

Methods

1. School selection-- On p. 5, the authors note that “four candidate secondary schools were selected in collaboration with local government officials.” It would be helpful to provide more detail on the criteria used for selecting these schools (e.g., size, location, existing mental health resources, or willingness to participate). Clarifying this would strengthen transparency around site selection and potential sources of selection bias.

2. Please clarify whether any incentives or compensation were provided to participants. If so, this information should be reported to ensure transparency and ethical clarity regarding participant engagement.

Intervention overview, p.7

1. Grammatical error on point number one, p.7, .. “including teachers and students and in charge of implementing TES activities.”

2. The statement on p.7—"Content included the core components described above…” may be misleading, as these components are only listed (not described) in the Introduction, which appears several pages earlier. Providing brief descriptors of each would enhance clarity and allow readers to better understand what these components entailed in practice. This level of detail is important for transparency, replication, and for readers less familiar with these specific intervention elements. It would strengthen clarity to

(a) briefly describe or summarize the key elements of these components here e.g. interpersonal skills (improving communication and relationship building; emotional regulation (identifying and managing emotions effectively etc.) and

(b) replace “described above” with “outlined in the Introduction” for accuracy and ease of reference.

3. How many hours did the sessions run for?

Selection of Facilitators p.8

1. While the manuscript provides helpful detail on the training and supervision facilitators received for implementation, it would be useful to also include brief information on their prior background or qualifications (e.g., whether they were teachers, counselors, or other professionals, and any prior mental health or psychosocial training). This context helps readers assess facilitator suitability and interpret results, especially given that intervention effects can vary by facilitator type in school-based settings. This would also support replication in similar school-based or low-resource contexts. Also include the number of facilitators, and approximately how many students in the TT groups to number of facilitator?

Sample size p.9

1. The authors note an aim to recruit 60 adolescents per school (n = 240 total) and explains that no formal sample size calculation was conducted due to the feasibility design. However, the actual number of participants enrolled, analyzed, and allocated to each trial arm is not clearly reported in the text; this information can only be inferred from the results table. For clarity and transparency, please state the final sample size per group (and any attrition, if applicable) directly in the Methods or Participants section. This will help readers assess recruitment success, allocation balance, and the interpretability of findings.

Process evaluation p.9-10

1. The authors mention that focus groups were conducted but does not provide key methodological details such as how many groups were held, with which participant categories, the number of participants per group, or the general topics or questions explored. Including this information would align with best practices for qualitative reporting and strengthen the transparency and credibility of the study’s qualitative component.

Quantitative assessments

1. The use of well-established and validated scales is a clear strength of this study and enhances the credibility of the findings. However, it would further strengthen the manuscript to report the reliability coefficients (e.g., Cronbach’s alpha) for these measures within the current sample. Providing this information helps assess how well the instruments performed in this specific context and supports confidence in the study’s results.

Qualitative analyses, p.11 bottom page

1. Please clarify what TM means or stands for that ‘thoroughly reviewed all transcripts twice.’

Results

1. Thriving Together p.12-13--Other than finding TT interesting, presented clearly, and feeling welcome, did students speak to its relation to mental health? It is unclear how these insights relate to the study’s core aims—particularly whether participants discussed how the intervention influenced their mental health understanding or outcomes, given that this is central to HASHTAG’s objectives. Providing greater clarity on the purpose and analytic focus of the focus groups, in earlier sections would help readers understand how these qualitative findings align with and inform the overall study

2. how participation in TT affected mental health understanding or outcomes given that is the goal of HASHTAG. Clarity on focus groups aims and intent would be helpful to know how results relate to what you were trying to assess/aims.

3. Outcome measures – It would strengthen the results section to include statistical evidence (e.g., coefficients, confidence intervals, and p-values) in the text for the reported significant within-group effects. Additionally, the absence of differences between intervention and control groups might partly relate to the measurement properties of the scales used. Since reliability for this sample is not reported, it is difficult to interpret whether measurement issues could have contributed to the non-significant findings. Reporting reliability coefficients would therefore be helpful for contextualizing these results.

Discussion

1. The Discussion section, line 3 notes “>80% completion rate of measures with adequate reliability,” suggesting that reliability was assessed and found acceptable. However, this information should be reported earlier—in the Measures section—so that readers can evaluate the psychometric quality of the instruments themselves and draw their own conclusions. Presenting reliability estimates transparently alongside the measure descriptions would strengthen methodological rigor and clarity.

2. The Discussion section is rich and provides valuable insights; however, it would benefit from being more explicitly linked to the study aims outlined earlier. For instance, the fifth aim was to pilot the outcome measures, yet there is limited discussion of the corresponding quantitative findings—why the results may have emerged as they did and what this implies for future measurement or intervention refinement. Clearly connecting each part of the discussion to the stated aims would improve coherence, strengthen interpretation, and help readers follow how the findings address the study’s objectives.

3. The manuscript would benefit from adding a dedicated limitations section, which is standard practice even for feasibility studies. For example, Table 1 shows that School B had one facilitator while School A had two, yet this discrepancy is not addressed in the text. On page 8, the authors refer to “each facilitator pair,” which seems inconsistent with the data presented. Discussing this and any other limitations—such as potential differences in implementation across schools or other contextual challenges—would enhance transparency and help readers interpret the findings more accurately.

---

## [Editor Report]

Dear Mr Mawoyo

Following peer review by two independent reviewer, the journal can now make a recommendation regarding your publication titled “Health Action in ScHools for a Thriving Adolescent Generation (HASHTAG): a feasibility trial of a school-based intervention for mental health promotion and prevention among adolescents in South Africa”, namely to re-submit with Major Revisions. In addition to the points of critique raised by the reviewers, kindly also elaborate on the definitions and associated measures for the concepts of feasibility and acceptability, by considering key resources in the implementation science field. Please address each critique raised and inidicate how these were resolved. We are looking forward to receiving your revised mauscript. Thank you for considering Cambridge Prisms: GMH. 

Best regards,

André J van Rensburg

---

## [Reviewer Report]

The authors have adequately addressed the minor revisions requested in the previous round. The manuscript is now clearly positioned as a feasibility study, with appropriate interpretation of quantitative and qualitative findings. The study makes a valuable contribution to the literature on school-based mental health interventions in LMIC settings. I recommend acceptance subject to minor editorial corrections.

---

## [Reviewer Report]

Thank you for your careful and thorough revisions. You have addressed all previously raised concerns, and the manuscript has been substantially strengthened as a result. I have no further major comments and believe the paper is now suitable for publication. I appreciate the authors’ responsiveness and clarity in the revisions.